# Is a Prestigious Job the same as a Prestigious Country?
# A Case Study on Multilingual Sentence Embeddings and European Countries

**Jindřich Libovický**

Institute of Formal and Applied Linguistics,
Faculty of Mathematics and Physics, Charles University, Czech Republic
`libovicky@ufal.mff.cuni.cz`

## Abstract

We study how multilingual sentence representations capture European countries and occupations and how this differs across European languages. We prompt the models with templated sentences that we machine-translate into 12 European languages and analyze the most prominent dimensions in the embeddings. Our analysis reveals that the most prominent feature in the embedding is the geopolitical distinction between Eastern and Western Europe and the country's economic strength in terms of GDP. When prompted specifically for job prestige, the embedding space clearly distinguishes high and low-prestige jobs. The occupational dimension is uncorrelated with the most dominant country dimensions in three out of four studied models. The exception is a small distilled model that exhibits a connection between occupational prestige and country of origin, which is a potential source of nationality-based discrimination. Our findings are consistent across languages.

## 1 Introduction

Language models and pre-trained representations in Natural Language Processing (NLP) are known to manifest biases against groups of people, including negative stereotypes connected to ethnicity or gender (Nangia et al., 2020; Nadeem et al., 2021). It has been extensively studied with monolingual models. Multilingual models, often used for model transfer between languages, introduce another potential issue: stereotypes of speakers of some languages can be imposed in other languages covered by the model.

In this case study, we try to determine the most prominent biases connected to European countries in multilingual sentence representation models. We adopt an unsupervised methodology (§ 2) based on hand-crafted prompt templates and principle component analysis (PCA), originally developed to extract moral sentiments from sentence representation (Schramowski et al., 2022).

Our exploration encompasses four sentence representation models across 13 languages (§ 3). We find only minor differences between languages in the models. The results (§ 4) show that the strongest dimension in all models correlates with the political and economic distinction between Western and Eastern Europe and the Gross Domestic Product (GDP). Prompting specifically for country prestige leads to similar results. When prompted for occupations, the models can distinguish between low and high-prestige jobs. In most cases, the extracted job-prestige dimension only loosely correlates with the country-prestige dimension. This result suggests that the models do not connect individual social prestige with the country of origin. The exception is a small model distilled from Multilingual Universal Sentence Encoder (Yang et al., 2020) that seems to mix these two and thus confirms previous work claiming that distilled models are more prone to biases (Ahn et al., 2022).

The source code for the experiments is available on GitHub.[1]

## 2 Methodology

We analyze sentence representation models (§ 2.1) using a generalization of the Moral Direction framework (§ 2.2). We represent concepts (countries, jobs) using sets of templated sentences (§ 2.3), for which we compute the sentence embeddings. Then, we compute the principal components of the embeddings and analyze what factors statistically explain the main principal component (§ 2.4).

### 2.1 Sentence Embeddings Models

Sentence-embedding models are trained to produce a single vector capturing the semantics of an en-

---

[1] https://github.com/jlibovicky/europe-in-sentence-embeddings

tire sentence. Contextual embeddings trained via the masked-language-modeling objective (Devlin et al., 2019) capture subwords well in context; however, they fail to provide a sentence representation directly comparable across sentences. Sentence-BERT (Reimers and Gurevych, 2019) approaches this problem by fine-tuning existing contextual embeddings using Siamese Networks on sentence classification tasks. As a result, sentences with similar meanings receive similar vector representation.

The issue of sentence representation also applies to multilingual contextual embeddings such as XLM-R (Conneau et al., 2020). In a multilingual setup, the additional requirement is that similar sentences receive similar representation regardless of the language. This is typically achieved using parallel data for knowledge distillation (Reimers and Gurevych, 2020; Heffernan et al., 2022) or more directly in a dual encoder setup (Feng et al., 2022).

## 2.2 Embedding Analysis Method

We base our methodology on an unsupervised method for extracting semantic dimensions from sentence embeddings, originally introduced in the context of moral institutions (Schramowski et al., 2022). The original study analyzed the moral sentiment of English verb phrases using SentenceBERT.

The method consists of three steps:

1. Generate templated sentences associating verbs with morality (e.g., "You should smile.", "It is good to smile.") and average them for each verb. I.e., there is one average sentence embedding per verb.

2. The sentences are processed with Sentence-BERT, and the representations are averaged for each phrase.

3. Apply PCA over the representations.

The results show that the most significant dimension roughly corresponds to the moral sentiment of the phrases. They use multiple templates so that linguistic differences and potential verb connotations average out. Using templated sentences also eliminates linguistic diversity in the data. Because of that, the main principle component does capture linguistic differences but the most prominent semantic nuances across the verbs when used in the specific context of moral intuitions.

We extend this method to a more exploratory setup. We use a similar set of template sentences, putting countries and occupations in the context of prestige. We average their embeddings and get the main principle component using PCA. Then, using three different template sets, we analyse what the main principle component best correlates with.

## 2.3 Templating Sentences

Similar to Hämmerl et al. (2023), who extended the Moral Dimension framework to multilingual models, we use templates in English and machine-translate the sentences into other languages after filling in the templates. We use three template sets. The sets consist of synonymous sentences with the following meaning:

1. They come from [COUNTRY].

2. Being from [COUNTRY] is considered prestigious.

3. Working as [JOB] is considered prestigious.

See Appendix A for the complete template list.

In the first set of templated sentences, we search for the general trend in how countries are represented. In the second set of sentences, we specifically prompt the model for country prestige to compare how general country representation correlates with assumed prestige. In the third case, we fit the PCA with templates containing job titles, i.e., the most prominent dimension captures job prestige according to the models. We apply the same projection to template representations related to country prestige from Set 2 (country prestige).

**Countries.** We include all European countries of the size of at least Luxembourg and use their short names (e.g., Germany instead of the Federal Republic of Germany), which totals 40 countries. The list of countries is in Appendix A.3.

**Low- and high-prestige jobs.** We base our list of low- and high-prestige jobs on a sociological study conducted in 2012 in the USA (Smith and Son, 2014). We manually selected 30 jobs for each category to avoid repetitions and to exclude US-specific positions. By using this survey, we also bring in the assumption that the European countries have approximately similar cultural distance from the USA. The complete list of job titles used is in Appendix A.2.

## 2.4 Evaluation

**Interpreting the dominant dimension.** For the analysis, we assign the countries with abstractive

| Model | Backbone | Parallel data | Params. |
|---|---|---|---|
| Mul. MPNet | XLM-R Base | Yes | 278M |
| D. mUSE | Distil-mBERT | No | 135M |
| LaBSE | — | Yes | 471M |
| XLM-R-NLI | XLM-R Base | No | 278M |

Table 1: Basic features of the studied models.

labels based on geographical (location, mountains, seas), political (international organization membership, common history), and linguistic features (see Table 5 in the Appendix for a detailed list). The labels are not part of the templates.

We compute the Pearson correlation of the one-hot indicator vector of the country labels with the extracted dominant dimension to provide an interpretation of the dimension (some also called point-biserial correlation). Finally, because creating a fixed unambiguous list of Western and Eastern countries is difficult and most criteria are ambiguous, we manually annotate if the most positively and negatively correlated labels correspond to the economic and political distinction between Eastern and Western Europe.

In addition, we compute the country dimension's correlation with the respective countries' GDP based on purchasing power parity in 2019, according to the World Bank.[2]

**Cross-lingual comparison.** We measure how the extracted dimensions correlate across languages. To explain where differences across languages come from, we compute how the differences correlate with the geographical distance of the countries where the languages are spoken, the GDP of the countries, and the lexical similarity of the languages (Bella et al., 2021).[3]

## 3 Experimental Setup

### 3.1 Evaluated Sentence Embeddings

We experimented with diverse sentence embedding models, which were trained using different methods. We experimented with models available in the SentenceBERT repository and an additional model. The overview of the models is in Table 1.

**Multilingual MPNet** was created by multilingual distillation from the monolingual English MPNet

Base model (Song et al., 2020) finetuned for sentence representation using paraphrasing (Reimers and Gurevych, 2019). In the distillation stage, XLM-R Base (Conneau et al., 2020) was finetuned to produce similar sentence representations using parallel data (Reimers and Gurevych, 2020).

**Distilled mUSE** is a distilled version of Multilingual Universal Sentence Encoder (Yang et al., 2020) that was distilled into Distill mBERT (Sanh et al., 2019). This model was both trained and distilled multilingually.

**LaBSE** (Feng et al., 2022) was trained on a combination of monolingual data and parallel data with a max-margin objective for better parallel sentence mining combined with masked language modeling.

**XLM-R-XNLI** is trained without parallel data using machine-translated NLI datasets (Hämmerl et al., 2023). The model is based on XLM-R Base but was finetuned using Arabic, Chinese, Czech, English, and German data following the Sentence-BERT recipe (Reimers and Gurevych, 2019).

### 3.2 Translating Templates

To evaluate the multilingual representations in more languages, we machine translate the templated text into 12 European languages: Bulgarian, Czech, German, Greek, Spanish, Finnish, French, Hungarian, Italian, Portuguese, Romanian, and Russian (and keep the English original). We selected languages for which high-quality machine translation systems are available on the Huggingface Hub. The models are listed in Appendix B.

## 4 Results

**Aggregated results.** The results aggregated over language are presented in Table 2. The detailed results per language are in the Appendix in Tables 6 and 7.

When prompting the models for countries, the most prominent dimensions almost always separate the countries according to the political east-west axis, consistently across languages. This is further stressed by the high correlation of the country dimension with the country's GDP, which is particularly strong in multilingual MPNet and Distilled mUSE. When we prompt the models specifically for country prestige, the correlation with the country's GDP slightly increases.

When we prompt the models for job prestige, they can distinguish high- and low-prestige jobs

---

[2]https://ourworldindata.org/grapher/gdp-per-capita-worldbank.

[3]http://ukc.disi.unitn.it/index.php/lexsim

| Model | Country of origin | | Country prestige | | Job prestige | | Job class. accuracy |
| --- | --- | --- | --- | --- | --- | --- | --- |
| | East-West | GDP cor. | East-West | GDP cor. | East-West | GDP cor. | |
| Multiling. Paraphrase MPNet | 1.00 | .79 | 1.00 | .79 | .08 | .08 | .93 |
| Distilled mUSE | 1.00 | .71 | 1.00 | .71 | .69 | .41 | .85 |
| LaBSE | 1.00 | .48 | 1.00 | .50 | .23 | .09 | .88 |
| NLI-finetuned XLM-R | .85 | .47 | .85 | .50 | .08 | .08 | .91 |

Table 2: Quantitative results averaged over languages showing the average correlation of the dominant dimension with the country's GDP and a proportion of languages where the dominant dimension corresponds to the political division of Eastern and Western countries. The detailed per-language results are presented in Tables 6 and 7 in the Appendix.

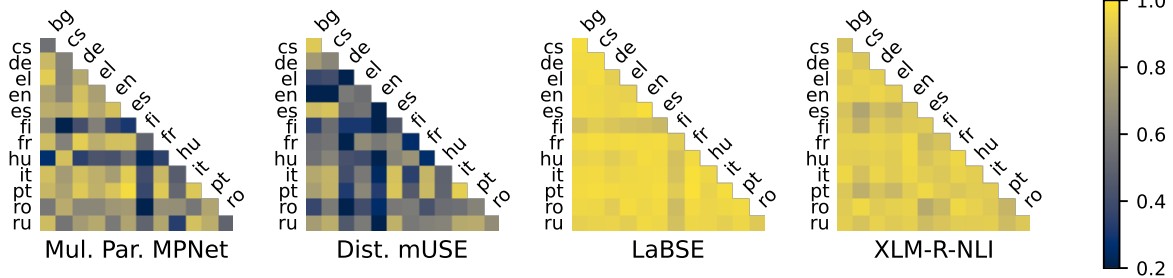

Figure 1: Cross-language correlation of the job-prestige dimension. Languages are coded using ISO 639-1 codes.

well (accuracy 85–93%). When we apply the same projection to prompts about countries, in most cases, the ordering of the countries is random. Therefore, we conclude that the models *do not* correlate job prestige and country of origin.

The only exception is Distilled mUSE, where the job-prestige dimension applied to countries still correlates with the country's GDP and the east-west axis. This is consistent with previous work showing that distilled student models exhibit more biases than model trained on authentic data (Vamvas and Sennrich, 2021; Ahn et al., 2022).

**Differences between languages.** Further, we evaluate how languages differ from each other.

In all models, the first PCA dimension from the job-prestige prompts separates low- and high-prestige jobs almost perfectly. Nevertheless, multilingual MPNet and distilled mUSE show a relatively low correlation of the job dimension across languages (see Figure 1).

Finally, we try to explain the correlation between languages by connecting them to countries where the languages are spoken. We measure how the correlation between languages correlates with the geographical distances of (the biggest) countries speaking the language, the difference in their GDP, and the lexical similarity of the languages. The results are presented in Table 3.

| Model | Geo. dist. | GDP diff. | Lang. sim. |
| --- | --- | --- | --- |
| MP MPNet | .020 | .077 | .459 |
| D. mUSE | .069 | .194 | .316 |
| LaBSE | .175 | .076 | .443 |
| XLM-R-NLI | .387 | .042 | .064 |

Table 3: Correlation of the language similarities (in terms of cross-language correlation of the job-prestige dimension) with the geographical distance of the countries, language similarity, and GDP.

For all models except XLM-R-NLI, the lexical similarity of the languages is the strongest predictor. XLM-R-NLI, with low differences between languages, better correlates with geographical distances.

## 5 Related Work

Societal biases of various types in neural NLP models are widely studied, especially focusing on gender and ethnicity. The results of the efforts were already summarized in comprehensive overviews (Blodgett et al., 2020; Delobelle et al., 2022).

Nationality bias has also been studied. Venkit et al. (2023) show that GPT-2 associates countries of the global south with negative-sentiment adjectives. However, only a few studies focus on biases in how multilingual models treat different lan-

guages. Papadimitriou et al. (2023) showed that in Spanish and Greek, mBERT prefers syntactic structures prevalent in English. Arora et al. (2022) and Hämmerl et al. (2023) studied differences in moral biases in multilingual language models, concluding there are some differences but no systematic trends. Yin et al. (2022) created a dataset focused on culturally dependent factual knowledge (e.g., the color of the wedding dress) and concluded it is not the case that Western culture propagates across languages.

## 6 Conclusions

We showed that all sentence representation models carry a bias that the most prominent feature of European countries is their economic power and affiliation to former geopolitical Western and Eastern blocks. In the models we studied, this presumed country prestige does not correlate with how the models represent the occupation status of people. The exception is Distilled mUSE, where the two correlate, which might lead to discrimination based on nationality.

## Limitations & Ethical Considerations

**The validity for different cultures.** The "ground truth" for job prestige was taken from studies conducted in the USA. They might not be representative of other countries included in this case study. Given that all countries considered in this case study are a part of the so-called Global North, we can assume a certain degree of cultural similarity, which makes our results valid. However, our methodology is *not guaranteed to generalize beyond the Western world*.

**Unintended use.** Some methods we use in the study might create a false impression that we have developed a scorer of job or country prestige. This is not the case. The correlations that we show in the Results section (§ 4) *do not guarantee the reliability of the scoring beyond the intended use* in the study, which is an assessment of multilingual sentence representation models. Drawing any conclusions about occupation groups, nations, or individual people using the methods used in this study might have harmful consequences.

## Acknowledgments

Many thanks to Tomáš Musil and Rudolf Rosa for discussing the methodology in this paper and to Ondřej Dušek and Dominik Macháček for their comments on the paper draft.

This research was supported by the Charles University project PRIMUS/23/SCI/023.

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

| Language | Code | Model |
|---|---|---|
| Bulgarian | bg | `Helsinki-NLP/opus-mt-tc-big-en-bg` |
| Czech | cs | `Helsinki-NLP/opus-mt-tc-big-en-ces_slk` |
| German | de | `facebook/wmt19-en-de` |
| Greek | el | `Helsinki-NLP/opus-mt-tc-big-en-el` |
| English | en | — |
| Spanish | es | `Helsinki-NLP/opus-mt-tc-big-en-es` |
| Finnish | fi | `Helsinki-NLP/opus-mt-tc-big-en-fi` |
| French | fr | `Helsinki-NLP/opus-mt-tc-big-en-fr` |
| Hungarian | hu | `Helsinki-NLP/opus-mt-tc-big-en-hu` |
| Italian | it | `Helsinki-NLP/opus-mt-tc-big-en-it` |
| Portuguese | pt | `Helsinki-NLP/opus-mt-tc-big-en-pt` |
| Romanian | ro | `Helsinki-NLP/opus-mt-tc-big-en-ro` |
| Russian | ru | `facebook/wmt19-en-ru` |

Table 4: Huggingface Hub Identifier of the machine translation models used for our experiments.

## A  Templated texts

### A.1  Sentence Templates

**Country of origin**

They are from *[COUNTRY]*.
They come from *[COUNTRY]*.
Their homeland in *[COUNTRY]*.
They are natives of *[COUNTRY]*.
They originate from *[COUNTRY]*.
They were born in *[COUNTRY]*.
Their country of birth is *[COUNTRY]*.
They hail from *[COUNTRY]*.
They grew up in *[COUNTRY]*.
They are citizens of *[COUNTRY]*.
*[COUNTRY]* is their country of origin.
*[COUNTRY]* is where they are from.
They were raised in *[COUNTRY]*.

#### A.1.1  Country prestige

Being from *[COUNTRY]* is considered prestigious.
Being from *[COUNTRY]* has high prestige.
Being from *[COUNTRY]* is seen as a symbol of success.
Being from *[COUNTRY]* is highly regarded.
People from *[COUNTRY]* are looked upon with admiration.
People from *[COUNTRY]* are highly esteemed.
It is prestigious to be from *[COUNTRY]*.

**Job prestige**

Being *[JOB]* is considered prestigious.
Being *[JOB]* has high prestige.
Working as *[JOB]* is seen as a symbol of success.
Working as *[JOB]* is highly regarded.
People who work as *[JOB]* are looked upon with admiration.
People who work as *[JOB]* are highly esteemed.
It is prestigious to work as *[JOB]*.

### A.2  Low- and High-prestige jobs

**Low-profile jobs.** a hotel chambermaid, a door-to-door salesman, a leaflet distributor, a janitor, a used car salesman, a bartender, a telephone operator, a carwash attendant, a cattle killer in a slaughtering plant, a dishwasher, a stockroom attendant, a box-folding-machine operator, a crushing-machine operator, a taxicab driver, a bicycle messenger, a salesperson in a hardware store, a street sweeper, a cashier in a supermarket, a pump operator, a railroad ticket agent, a desk clerk in a hotel, a cable TV installer, a sewing machine operator, a waiter in a restaurant, an assembly line worker, a shoeshiner, a ditch digger, an unskilled worker in a factory, a tire retreader, a dry cleaner

**High-profile jobs.** a surgeon, a university professor, an architect, a lawyer, a priest, a banker, a school principal, an airline pilot, an economist, a network administrator, an air traffic controller, an author, a nuclear plant operator, a computer scientist, a psychologist, a pharmacist, a colonel in the army, a mayor of a city, a university president, a dentist, a fire department lieutenant, a high school teacher, a policeman, a software developer, an actor, a fashion model, a journalist, a musician in a symphony orchestra, a psychiatrist, a chemical engineer

### A.3  Countries

We consider the following 40 countries (ordered by their ISO 3166-1 codes): Austria, Bosnia and Herzegovina, Belgium, Bulgaria, Belarus, Switzerland, the Czech Republic, Cyprus, Den-

mark, Germany, Greece, Spain, Estonia, Finland, France, Hungary, Croatia, Ireland, Iceland, Italy, Latvia, Lithuania, Luxembourg, Netherlands, Moldova, Montenegro, North Macedonia, Malta, Norway, Portugal, Poland, Romania, Russia, Slovakia, Slovenia, Albania, Serbia, Sweden, Turkey, Ukraine, Great Britain, Kosovo.

The qualitative group labels we assign to the countries we use in the further analysis are in Table 5. The values reflect the world as in the training data (estimated pre-2021) for the models and, therefore, do not reflect recent events (i.e., Croatia is not listed among countries paying with Euro, and Finland is considered neutral).

## B  Machine Translation Models

The machine translation models we used are listed in Table 4 While keeping default values for all decoding parameters.

## C  Detailed per-language results

The detailed per-language results are presented in Tables 6 and 7.

| Label | Austria | Bosnia and Herzegovina | Belgium | Bulgaria | Belarus | Switzerland | Czech Republic | Cyprus | Denmark | Germany | Greece | Spain | Estonia | Finland | France | Hungary | Croatia | Ireland | Iceland | Italy | Latvia | Lithuania | Luxembourg | Netherlands | Moldova | Montenegro | North Macedonia | Malta | Norway | Portugal | Poland | Romania | Russia | Slovakia | Slovenia | Albania | Serbia | Sweden | Turkey | Ukraine | Great Britain | Kosovo |
|---|---|---|---|---|---|---|---|---|---|---|---|---|---|---|---|---|---|---|---|---|---|---|---|---|---|---|---|---|---|---|---|---|---|---|---|---|---|---|---|---|---|---|
| *Geographical* | | | | | | | | | | | | | | | | | | | | | | | | | | | | | | | | | | | | | | | | | | |
| Alps | ✓ | | | | | ✓ | | | | ✓ | | | | | ✓ | | | | | ✓ | | | | | | | | | | | | | | | ✓ | | | | | | | |
| Arctic Sea | | | | | | | | | | | | | | | | | | | | | | | | | | | | | ✓ | | | | ✓ | | | | | | | | | |
| Atlantic Ocean | | | | | | | | | | | | ✓ | | | ✓ | | | ✓ | ✓ | | | | | | | | | | ✓ | ✓ | | | | | | | | | | | ✓ | |
| Balkan Peninsula | ✓ | ✓ | | ✓ | | | | | | | ✓ | | | | | | ✓ | | | | | | | | | ✓ | ✓ | | | | | | | | ✓ | ✓ | ✓ | | | | | ✓ |
| Baltic Sea | | | | | | | | | ✓ | ✓ | | | ✓ | ✓ | | | | | | | ✓ | ✓ | | | | | | | | | ✓ | | ✓ | | | | | ✓ | | | | |
| Big Country | | | | | | | | | | ✓ | | ✓ | | | ✓ | | | | | ✓ | | | | | | | | | | | ✓ | | | | | | | | | ✓ | | |
| Black Sea | | | | ✓ | | | | | | | | | | | | | | | | | | | | | | | | | | | | ✓ | ✓ | | | | | | ✓ | ✓ | | |
| Central Europe | ✓ | | | | | | ✓ | | | ✓ | | | | | | | | | | | | | | | | | | | | | | | | | | | | | | | | |
| Coastal State | ✓ | ✓ | ✓ | | | | | | ✓ | ✓ | ✓ | ✓ | ✓ | ✓ | ✓ | | ✓ | ✓ | | ✓ | ✓ | ✓ | | ✓ | | ✓ | | ✓ | ✓ | | | | | | | | | | | | | |
| Eastern Europe | | | | ✓ | ✓ | | | | | | | | ✓ | | | | | | | | ✓ | ✓ | | | ✓ | | | | | | ✓ | ✓ | ✓ | ✓ | | | | | | ✓ | | |
| Island | | | | | | | | ✓ | | | | | | | | | | | ✓ | | | | | | | | | ✓ | | | | | | | | | | | | | | ✓ |
| Landlocked | ✓ | | | | | ✓ | ✓ | | | | | | | | | ✓ | | | | | | | ✓ | | ✓ | | ✓ | | | | | | | ✓ | | | ✓ | | | | | ✓ |
| Mediterranean Sea | | | | | | | | ✓ | | | ✓ | ✓ | | | ✓ | | ✓ | | | ✓ | | | | | | ✓ | | ✓ | | | | | | | ✓ | ✓ | | | ✓ | | | |
| Northern Europe | | | | | | | | | ✓ | | | | ✓ | ✓ | | | | | ✓ | | | | | | | | | | ✓ | | | | | | | | | ✓ | | | | |
| North Sea | | | ✓ | | | | | | ✓ | ✓ | | | | | | | | | | | | | | ✓ | | | | | ✓ | | | | | | | | | | | | ✓ | |
| Southern Europe | ✓ | | | | | | | | | | ✓ | ✓ | | | | | | | | ✓ | | | | | | ✓ | ✓ | ✓ | | ✓ | | | | | ✓ | ✓ | ✓ | | ✓ | | | ✓ |
| South-East Europe | ✓ | ✓ | | | | | | | | | | | | | | | | | | | | | | | | | | | | | | | | | | | | | | | | ✓ |
| Western Europe | ✓ | | ✓ | | | ✓ | | | | ✓ | | | | | ✓ | | | ✓ | | | | | ✓ | ✓ | | | | | | | | | | | | | | | | | ✓ | |
| *Linguistic* | | | | | | | | | | | | | | | | | | | | | | | | | | | | | | | | | | | | | | | | | | |
| Baltic Family | | | | | | | | | | | | | | | | | | | | | ✓ | ✓ | | | | | | | | | | | | | | | | | | | | |
| Cyrillic Alphabet | | | | ✓ | ✓ | | | | | | | | | | | | | | | | | | | | | | | | | | | | ✓ | | | | ✓ | | | ✓ | | |
| English-speaking | | | | | | | | | | | | | | | | | | ✓ | | | | | | | | | | ✓ | | | | | | | | | | | | | ✓ | |
| French-speaking | | | ✓ | | | ✓ | | | | | | | | | ✓ | | | | | | | | ✓ | | | | | | | | | | | | | | | | | | | |
| German-speaking | ✓ | | ✓ | | | ✓ | | | | ✓ | | | | | | | | | | | | | ✓ | | | | | | | | | | | | | | | | | | | |
| Germanic Family | ✓ | | ✓ | | | ✓ | | | ✓ | ✓ | | | | | | | | ✓ | ✓ | | | | ✓ | ✓ | | | | | ✓ | | | | | | | | | ✓ | | | ✓ | |
| Non-Latin Script | | | | ✓ | ✓ | | | | | | ✓ | | | | | | | | | | | | | | | | | | | | | | ✓ | | | | ✓ | | | ✓ | | |
| Romance Family | | | ✓ | | | ✓ | | | | | | ✓ | | | ✓ | | | | | ✓ | | | | | ✓ | | | | | ✓ | | ✓ | | | | | | | | | | |
| Slavic Family | ✓ | ✓ | | ✓ | ✓ | | ✓ | | | | | | | | | | ✓ | | | | | | | | | ✓ | ✓ | | | | ✓ | | ✓ | ✓ | ✓ | | ✓ | | | ✓ | | |
| Ugro-Finnic Family | | | | | | | | | | | | | ✓ | ✓ | | ✓ | | | | | | | | | | | | | | | | | | | | | | | | | | |
| *Political* | | | | | | | | | | | | | | | | | | | | | | | | | | | | | | | | | | | | | | | | | | |
| Baltic States | | | | | | | | | | | | | ✓ | | | | | | | | ✓ | ✓ | | | | | | | | | | | | | | | | | | | | |
| Benelux | | | ✓ | | | | | | | | | | | | | | | | | | | | ✓ | ✓ | | | | | | | | | | | | | | | | | | |
| EU member | ✓ | | ✓ | ✓ | | | ✓ | ✓ | ✓ | ✓ | ✓ | ✓ | ✓ | ✓ | ✓ | ✓ | ✓ | ✓ | | ✓ | ✓ | ✓ | ✓ | ✓ | | | | ✓ | | ✓ | ✓ | ✓ | | ✓ | ✓ | | | ✓ | | | | |
| New EU Member | | | | ✓ | | | ✓ | ✓ | | | | | ✓ | | | ✓ | ✓ | | | | ✓ | ✓ | | | | | | ✓ | | | ✓ | ✓ | | ✓ | ✓ | | | | | | | |
| EU-15 Member | ✓ | | ✓ | | | | | | ✓ | ✓ | ✓ | ✓ | | ✓ | ✓ | | | ✓ | | ✓ | | | ✓ | ✓ | | | | | | ✓ | | | | | | | | ✓ | | | ✓ | |
| Euro as currency | | | ✓ | | | | | ✓ | | ✓ | ✓ | ✓ | ✓ | ✓ | ✓ | | | ✓ | | ✓ | ✓ | ✓ | ✓ | ✓ | | | | ✓ | | ✓ | | | | ✓ | ✓ | | | | | | | |
| G7 member | | | | | | | | | | ✓ | | | | | ✓ | | | | | ✓ | | | | | | | | | | | | | | | | | | | | | ✓ | |
| Monarchy | | | ✓ | | | | | | ✓ | | | ✓ | | | | | | | | | | | ✓ | ✓ | | | | | ✓ | | | | | | | | | ✓ | | | ✓ | |
| NATO member | | ✓ | ✓ | ✓ | | | ✓ | | ✓ | ✓ | ✓ | ✓ | ✓ | | ✓ | ✓ | ✓ | | ✓ | ✓ | ✓ | ✓ | ✓ | ✓ | | ✓ | ✓ | | ✓ | ✓ | ✓ | ✓ | | ✓ | ✓ | ✓ | | | ✓ | | ✓ | |
| Neutral | ✓ | | | | | ✓ | | | | | | | | ✓ | | | | ✓ | | | | | | | | | | | | | | | | | | | | ✓ | | | | |
| Former Yugoslavia | | ✓ | | | | | | | | | | | | | | | ✓ | | | | | | | | | ✓ | ✓ | | | | | | | | ✓ | | ✓ | | | | | ✓ |
| Post-communist | ✓ | | | ✓ | ✓ | | ✓ | | | | | | ✓ | | | ✓ | ✓ | | | | ✓ | ✓ | | | ✓ | ✓ | ✓ | | | | ✓ | ✓ | ✓ | ✓ | ✓ | ✓ | ✓ | | | ✓ | | ✓ |
| Post-soviet | | | | | ✓ | | | | | | | | ✓ | | | | | | | | ✓ | ✓ | | | ✓ | | | | | | | | ✓ | | | | | | | ✓ | | |
| Schengen Area | | | ✓ | | | ✓ | ✓ | | ✓ | | | | | | | | | | | ✓ | ✓ | ✓ | ✓ | ✓ | | | | | ✓ | ✓ | ✓ | | | | | | | | | | | |
| Visegrad Four | | | | | | | ✓ | | | | | | | | | ✓ | | | | | | | | | | | | | | | ✓ | | | ✓ | | | | | | | | |

Table 5: Labels for grouping countries.

Multilingual Paraphrase MPNet (paraphrase-multilingual-mpnet-base-v2)

| Lng | Country of origin | | | | | | Country prestige | | | | | | Job prestige applied to country | | | | | | Job class. acc. |
|---|---|---|---|---|---|---|---|---|---|---|---|---|---|---|---|---|---|---|---|
| | The main direction | | | corr. | Is east-west? | Corr. w/ GDP | The main direction | | | corr. | Is east-west? | Corr. w/ GDP | The main direction | | | corr. | Is east-west? | Corr. w/ GDP | |
| | ⊖ label | corr. | ⊕ label | | | | ⊖ label | corr. | ⊕ label | | | | ⊖ label | corr. | ⊕ label | | | | |
| bg | Slavic lng. | -.70 | West | .70 | ✓ | .80 | Slavic lng. | -.71 | West | .70 | ✓ | .80 | North | -.31 | German | .50 | — | — | .93 |
| cs | Slavic lng. | -.69 | West | .71 | ✓ | .81 | Slavic lng. | -.69 | West | .71 | ✓ | .81 | North | -.37 | Baltic state | .43 | — | — | .93 |
| de | Slavic lng. | -.69 | West | .68 | ✓ | .80 | Balkan | -.70 | Germanic lng. | .69 | ✓ | .81 | North | -.38 | German | .53 | — | — | .93 |
| el | Slavic lng. | -.70 | West | .68 | ✓ | .80 | Slavic lng. | -.70 | West | .70 | ✓ | .81 | North | -.44 | German | .46 | — | — | .90 |
| en | Slavic lng. | -.69 | West | .71 | ✓ | .80 | Slavic lng. | -.70 | West | .71 | ✓ | .80 | North | -.43 | German | .51 | — | .33 | .97 |
| es | Slavic lng. | -.69 | West | .71 | ✓ | .80 | Slavic lng. | -.68 | West | .72 | ✓ | .81 | North | -.42 | German | .48 | ✓ | — | .93 |
| fi | Balkan | -.67 | West | .71 | ✓ | .81 | Balkan | -.70 | West | .68 | ✓ | .78 | Post-Soviet | .32 | German | .50 | — | — | .93 |
| fr | Slavic lng. | -.67 | West | .71 | ✓ | .81 | Balkan | -.68 | West | .69 | ✓ | .81 | North | -.36 | German | .49 | — | — | .90 |
| hu | Slavic lng. | -.69 | West | .72 | ✓ | .80 | Slavic lng. | -.70 | West | .71 | ✓ | .80 | North | -.57 | Post-Soviet | .31 | — | — | .95 |
| it | Slavic lng. | -.69 | West | .71 | ✓ | .81 | Slavic lng. | -.71 | West | .69 | ✓ | .81 | North | -.36 | German | .53 | — | .39 | .97 |
| pt | Slavic lng. | -.67 | West | .71 | ✓ | .80 | Slavic lng. | -.68 | West | .70 | ✓ | .81 | North | -.47 | German | .45 | — | — | .93 |
| ro | Germanic lng. | -.67 | Slavic lng. | .72 | ✓ | .81 | Slavic lng. | -.71 | Germanic lng. | .68 | ✓ | .81 | North | -.38 | German | .46 | — | .31 | .93 |
| ru | Post-comm. | -.71 | West | .63 | ✓ | .62 | Post-comm. | -.69 | West | .62 | ✓ | .63 | Monarchy | -.36 | Alps | .48 | — | — | .92 |
| Average | | -.69 | | .70 | 1 | .79 | | -.70 | | .69 | 1 | .79 | | -.35 | | .47 | 0.08 | .08 | .93 |

Distilled Multilingual Sentence Encoder (distiluse-base-multilingual-cased-v2)

| Lng | Country of origin | | | | | | Country prestige | | | | | | Job prestige applied to country | | | | | | Job class. acc. |
|---|---|---|---|---|---|---|---|---|---|---|---|---|---|---|---|---|---|---|---|
| | The main direction | | | corr. | Is east-west? | Corr. w/ GDP | The main direction | | | corr. | Is east-west? | Corr. w/ GDP | The main direction | | | corr. | Is east-west? | Corr. w/ GDP | |
| | ⊖ label | corr. | ⊕ label | | | | ⊖ label | corr. | ⊕ label | | | | ⊖ label | corr. | ⊕ label | | | | |
| bg | Germanic lng. | -.66 | Post-comm. | .61 | ✓ | .73 | Post-comm. | -.61 | Germanic lng. | .67 | ✓ | .73 | German | -.54 | Yugoslavia | .54 | ✓ | .63 | .80 |
| cs | Post-comm. | -.61 | Germanic lng. | .65 | ✓ | .72 | Post-comm. | -.61 | Germanic lng. | .65 | ✓ | .71 | Yugoslavia | -.42 | EU-15 | .52 | ✓ | .38 | .90 |
| de | Germanic lng. | -.59 | Slavic lng. | .63 | ✓ | .68 | Germanic lng. | -.60 | Post-comm. | .69 | ✓ | .73 | Post-comm. | -.47 | Germanic lng. | .48 | ✓ | .65 | .88 |
| el | Post-comm. | -.64 | Germanic lng. | .62 | ✓ | .71 | Post-comm. | -.63 | Germanic lng. | .62 | ✓ | .70 | North | -.34 | Post-Soviet | .40 | — | — | .82 |
| en | Germanic lng. | -.62 | Slavic lng. | .63 | ✓ | .71 | Germanic lng. | -.63 | Slavic lng. | .62 | ✓ | .72 | Yugoslavia | -.53 | EU-15 | .49 | ✓ | .62 | .87 |
| es | Germanic lng. | -.65 | Slavic lng. | .63 | ✓ | .74 | Germanic lng. | -.66 | Post-comm. | .63 | ✓ | .76 | Yugoslavia | -.53 | EU-15 | .50 | ✓ | .64 | .83 |
| fi | West | -.61 | Balkan | .64 | ✓ | .72 | Balkan | -.61 | West | .56 | ✓ | .70 | Neutral | .31 | EU | .43 | — | — | .78 |
| fr | Germanic lng. | -.57 | Post-comm. | .63 | ✓ | .67 | EU-15 | -.58 | Post-comm. | .64 | ✓ | .68 | Yugoslavia | -.55 | EU-15 | .46 | ✓ | .58 | .88 |
| hu | Post-comm. | -.63 | Germanic lng. | .61 | ✓ | .66 | Post-comm. | -.64 | Germanic lng. | .63 | ✓ | .68 | Post-comm. | -.48 | EU-15 | .46 | ✓ | .37 | .83 |
| it | Germanic lng. | -.65 | Slavic lng. | .62 | ✓ | .74 | Post-comm. | -.61 | Germanic lng. | .63 | ✓ | .72 | Yugoslavia | -.36 | Atlantic | .33 | — | .32 | .87 |
| pt | Balkan | -.58 | Germanic lng. | .69 | ✓ | .74 | Germanic lng. | -.58 | Germanic lng. | .69 | ✓ | .73 | Yugoslavia | -.46 | EU-15 | .47 | ✓ | .54 | .82 |
| ro | Germanic lng. | -.62 | Post-comm. | .64 | ✓ | .70 | Germanic lng. | -.64 | Post-comm. | .64 | ✓ | .72 | EU-15 | -.51 | Yugoslavia | .57 | ✓ | .57 | .83 |
| ru | Post-comm. | -.63 | Germanic lng. | .61 | ✓ | .65 | Post-comm. | -.60 | Germanic lng. | .58 | ✓ | .60 | Yugoslavia | -.53 | alps | .34 | — | — | .88 |
| Average | | -.62 | | .63 | 1 | .70 | | -.61 | | .63 | 1 | .71 | | -.42 | | .46 | 0.69 | .41 | .85 |

Table 6: Detailed per-language results for Multilingual MPNet and Distilled Multilingual Sentence Encoder.

**LaBSE** (`sentence-transformers/LaBSE`)

| | Country of origin | | | | | | Country prestige | | | | | | Job prestige applied to country | | | | | | Job class. acc. |
|---|---|---|---|---|---|---|---|---|---|---|---|---|---|---|---|---|---|---|---|
| Lng | ⊖ label | corr. | ⊕ label | corr. | Is east-west? | Corr. w/GDP | ⊖ label | corr. | ⊕ label | corr. | Is east-west? | Corr. w/GDP | ⊖ label | corr. | ⊕ label | corr. | Is east-west? | Corr. w/GDP | |
| bg | eu15 | -.37 | Yugoslavia | .59 | ✓ | .48 | Euro | -.36 | Balkan | .53 | ✓ | .45 | Atlantic | -.37 | Yugoslavia | .36 | — | — | .83 |
| cs | North | -.48 | Slavic lng. | .63 | ✓ | .51 | eu15 | -.49 | Slavic lng. | .62 | ✓ | .54 | Baltic lng. | -.42 | Atlantic | .48 | — | — | .90 |
| de | Atlantic | -.44 | Slavic lng. | .61 | ✓ | .45 | Germanic lng. | -.43 | Slavic lng. | .51 | ✓ | .58 | Big | -.43 | Yugoslavia | .35 | — | — | .95 |
| el | Slavic lng. | -.51 | eu15 | .51 | ✓ | .45 | Slavic lng. | -.50 | eu15 | .53 | ✓ | .46 | South-East | -.46 | Big | .41 | — | .39 | .85 |
| en | Euro | -.37 | Yugoslavia | .59 | ✓ | .42 | eu15 | -.43 | Yugoslavia | .63 | ✓ | .43 | South | -.38 | Yugoslavia | .38 | — | — | .83 |
| es | eu15 | -.47 | Slavic lng. | .64 | ✓ | .56 | eu15 | -.53 | Slavic lng. | .61 | ✓ | .57 | South | -.36 | Yugoslavia | .37 | — | — | .88 |
| fi | Balkan | -.62 | Germanic lng. | .44 | ✓ | .55 | Germanic lng. | -.46 | Balkan | .62 | ✓ | .57 | Landlocked | -.40 | Atlantic | .45 | — | — | .92 |
| fr | North | -.49 | Slavic lng. | .62 | ✓ | .52 | EU-15 | -.58 | Slavic lng. | .63 | ✓ | .56 | EU-15 | -.47 | Yugoslavia | .49 | ✓ | .40 | .87 |
| hu | North | -.45 | Balkan | .61 | ✓ | .51 | EU-15 | -.48 | Balkan | .58 | ✓ | .52 | Neutral | -.33 | Neutral | -.31 | — | — | .85 |
| it | eu15 | -.48 | Slavic lng. | .59 | ✓ | .47 | EU-15 | -.50 | Slavic lng. | .58 | ✓ | .47 | Atlantic | -.35 | Yugoslavia | .40 | ✓ | — | .87 |
| pt | North | -.49 | Slavic lng. | .60 | ✓ | .51 | Slavic lng. | -.59 | North | .51 | ✓ | .52 | EU | -.37 | Yugoslavia | .37 | — | — | .90 |
| ro | Slavic lng. | -.56 | North | .43 | ✓ | .45 | Slavic lng. | -.58 | eu15 | .45 | ✓ | .44 | Non-Latin | -.31 | Schengen | .38 | — | — | .87 |
| ru | North | -.36 | South-East | .54 | ✓ | .41 | Neutral | -.38 | Yugoslavia | .51 | ✓ | .38 | EU-15 | -.40 | Yugoslavia | .71 | ✓ | .33 | .88 |
| | | -.47 | | .57 | 1 | .48 | | -.49 | | .56 | 1 | .50 | | -.39 | | .37 | 0.23 | .09 | .88 |

**XLM-R finetuned on NLI**

| | Country of origin | | | | | | Country prestige | | | | | | Job prestige applied to country | | | | | | Job class. acc. |
|---|---|---|---|---|---|---|---|---|---|---|---|---|---|---|---|---|---|---|---|---|
| Lng | ⊖ label | corr. | ⊕ label | corr. | Is east-west? | Corr. w/GDP | ⊖ label | corr. | ⊕ label | corr. | Is east-west? | Corr. w/GDP | ⊖ label | corr. | ⊕ label | corr. | Is east-west? | Corr. w/GDP | |
| bg | Atlantic | -.51 | Yugoslavia | .71 | ✓ | .60 | Slavic lng. | -.67 | EU-15 | .59 | ✓ | .64 | east | -.42 | Germanic lng. | .48 | ✓ | .57 | .90 |
| cs | Neutral | -.40 | Benelux | .35 | ✓ | — | North-sea | -.45 | Slavic lng. | .33 | ✓ | .32 | — | — | — | — | — | — | .85 |
| de | EU-15 | -.47 | Slavic lng. | .59 | ✓ | .55 | EU-15 | -.52 | Post-comm. | .64 | ✓ | .52 | Baltic state | -.46 | Balkan | .35 | — | — | .92 |
| el | west | -.40 | Slavic lng. | .42 | ✓ | .35 | west | -.44 | Slavic lng. | .46 | ✓ | .36 | South | -.40 | Post-Soviet | .41 | — | — | .87 |
| en | Balkan | -.52 | EU-15 | .64 | ✓ | .57 | Post-comm. | -.62 | EU-15 | .73 | ✓ | .66 | Visegrad | -.32 | Romance lng. | -.31 | — | — | .97 |
| es | Euro | -.56 | Cyrillic | .51 | ✓ | .45 | Euro | -.56 | Post-comm. | .60 | ✓ | .46 | Neutral | -.38 | Island | .44 | — | — | .98 |
| fi | Post-comm. | -.56 | Atlantic | .61 | ✓ | .58 | Post-comm. | -.58 | Atlantic | .61 | ✓ | .65 | South | -.37 | South | -.37 | — | — | .92 |
| fr | coastal | -.42 | Post-comm. | .49 | — | — | coastal | -.45 | Post-comm. | .46 | — | — | romance-lng | -.36 | Visegrad | .32 | — | — | .83 |
| hu | Euro | -.44 | Slavic lng. | .50 | ✓ | .39 | NATO | -.54 | Slavic lng. | .50 | ✓ | .41 | Black Sea | -.31 | Visegrad | .41 | — | — | .92 |
| it | Euro | -.45 | Cyrillic | .49 | ✓ | — | English | -.32 | Slavic lng. | .57 | — | — | Romance lng. | -.42 | German | .34 | — | .41 | .93 |
| pt | Balkan | -.36 | Neutral | .57 | ✓ | .34 | Balkan | -.45 | Neutral | .59 | — | .51 | South | -.50 | Baltic | .44 | — | — | .92 |
| ro | Euro | -.44 | Slavic lng. | .51 | ✓ | .44 | Atlantic | -.51 | Slavic lng. | .58 | ✓ | .53 | — | — | — | — | — | — | .85 |
| ru | Slavic lng. | -.58 | Atlantic | .47 | ✓ | .44 | Slavic lng. | -.55 | North sea | .48 | ✓ | .42 | South | -.39 | Yugoslavia | -.32 | — | — | .92 |
| | | -.47 | | .53 | 0.85 | .47 | | -.51 | | .55 | 0.85 | .50 | | -.39 | | .20 | 0.08 | .08 | .91 |

Table 7: Detailed per-language results for LaBSE and XLM-R finetuned on NLI.