# OpenReview forum: "Is a Prestigious Job the same as a Prestigious Country? A Case Study on Multilingual Sentence Embeddings and European Countries"
_EMNLP/2023/Conference — EMNLP 2023 Findings_

### Official Review · Reviewer_QesP · 2023-07-31

**Soundness:** 3

**Excitement:**

4: Strong: This paper deepens the understanding of some phenomenon or lowers the barriers to an existing research direction.

**Paper Topic And Main Contributions:**

This paper aims to analyze the biases connected to European countries in multilingual sentence representation models. Specifically:

* The strongest dimension in all models correlates with the political and economic distinction between Western and Eastern Europe and the GDP.
* The models can distinguish between low and high-prestige jobs.
* Except for one model, all models only loosely associate job prestige with country prestige.

**Questions For The Authors:**

1. Why did you choose to machine translate, instead of using existing data from the corpora, like EuroParl?
2. Would it make sense for you to elaborate on the potential ethical concerns based on your findings?
3. How were the templates and models chosen? Could you briefly explain your rationale behind it?


**Reasons To Accept:**

This paper investigates some of the most important questions in our field. Knowing the biases in our model and data helps us make the better decision.

**Reasons To Reject:**

The scope of this study is relatively small, even for a focus contribution, thus making the results not as reliable.

**Reproducibility:**

4: Could mostly reproduce the results, but there may be some variation because of sample variance or minor variations in their interpretation of the protocol or method.

**Reviewer Confidence:**

4: Quite sure. I tried to check the important points carefully. It's unlikely, though conceivable, that I missed something that should affect my ratings.

**Typos Grammar Style And Presentation Improvements:**

The graphs in Fig. 1 are a little less intuitive.
Expanding the scope of this study and making it a long paper would be great!

---

> ### Author Rebuttal · Authors · 2023-08-24
>
> **Why MT.** The method we use requires only minimum semantic differences between the sentences, e.g., in the country sentences, we want the only word that changes to be the country name. Otherwise, as previous work showed, the most prominent PCA dimensions would capture linguistic features. Using authentic parallel sentences would introduce too much linguistic diversity for the method to work well. Therefore, we use templated sentences that we machine-translate into multiple languages.
>
> **Ethical concerns.** As mentioned in the response for Reviewer 3DFr, the application areas we had in mind when designing the experiments were problematic NLP applications, such as automated processing of CVs with potentially harmful consequences for individuals. Moreover, when people attribute importance to economic and geopolitical factors (especially when thinking about individuals), in some sense, it reflects their moral values. However, to what extent the same can be said about sentence embeddings is questionable, and we did not feel qualified to discuss this topic in detail.
>
> **Template design.** We want the sentences to capture the meaning as simply as possible. The templates should be structurally simple, so their linguistic features do not influence the sentence embedding. Also, simple sentences get better translated using MT. We use a set of synonymous templates for each category so that potential meaning nuances average out in the embedding space.

---

### Official Review · Reviewer_os6q · 2023-08-05

**Soundness:** 3

**Excitement:**

3: Ambivalent: It has merits (e.g., it reports state-of-the-art results, the idea is nice), but there are key weaknesses (e.g., it describes incremental work), and it can significantly benefit from another round of revision. However, I won't object to accepting it if my co-reviewers champion it.

**Paper Topic And Main Contributions:**

This is an exploratory paper that studies and exposes how pre-trained multilingual language models inherently contain bias or make distinctions among European countries. The authors show a reflection of geopolitics (eastern vs Western European countries) and economy based (in terms of GDP) distinctions. They relied on hand-crafted templates and PCA to showcase their findings. PCA finds the strongest dimensions across which sentence embeddings corresponding to the templates are placed, and the dimension values correlate with political and economic distinction among Western and eastern European countries. For job profiles, models show a clear distinction between high and low-profile jobs; however, the majority of them do not show a strong correlation between job prestige dimensions and country dimensions.

**Reasons To Accept:**

Timely study. Interest to the broader NLP community.

Several results are fascinating. Very few such studies have been done on multilingual embeddings in the past.

**Reasons To Reject:**

There are no mathematical expressions showcasing the operational steps. This makes the paper extremely difficult to understand. A reader has to entirely rely on intuition to understand the inherent processes.

The entire 2.4 section needs to be elaborated clearly.

Not sure which correlation the authors have used

**Reproducibility:**

3: Could reproduce the results with some difficulty. The settings of parameters are underspecified or subjectively determined; the training/evaluation data are not widely available.

**Reviewer Confidence:**

3: Pretty sure, but there's a chance I missed something. Although I have a good feel for this area in general, I did not carefully check the paper's details, e.g., the math, experimental design, or novelty.

---

> ### Author Rebuttal · Authors · 2023-08-24
>
> **Missing formulas.** Thank you for pointing out less clear points in our writing. We decided not to write formulas to save space in a 4-page short paper. We can address the writing issues in the camera-ready version with the extra page.
>
> **Correlation.** Every time we mention correlation, it is the Pearson correlation coefficient. In Section 2.4, we talk about the Person correlation between the extracted most prominent (PCA) dimension from the embedding with the indicator vectors for country features (i.e., one if the country has the quality, zero if it does not; e.g., if the country uses Euro as its currency). This is sometimes called Point-biserial correlation.

---

### Official Review · Reviewer_3DFr · 2023-08-05

**Soundness:** 3

**Excitement:**

3: Ambivalent: It has merits (e.g., it reports state-of-the-art results, the idea is nice), but there are key weaknesses (e.g., it describes incremental work), and it can significantly benefit from another round of revision. However, I won't object to accepting it if my co-reviewers champion it.

**Paper Topic And Main Contributions:**

In this work, the authors show that several sentence embedding models present social biases in how country names and jobs are encoded. In particular, a set of metadata features are manually associated with top PCA components for investigation based on correlation analysis. Experimental results show high correlation between top PCA component with two features: Western and Eastern Europe economic distinction and Gross Domestic Product. Low correlation is then observed for jobs prestige. These observations are consistent across multiple languages.

**Reasons To Accept:**

- Sound case study on bias analysis covering multiple languages

**Reasons To Reject:**

- Unclear impact of the proposed study

## Recommendation

The proposed case study is interesting and sound. Moreover, the paper is well organized and written. My main concern is about the impact and purpose of such study, which appears to be not sufficiently remarked in the paper. Any dataset has some biases since it is a limited collection of data [1]. The question is whether this type of bias is (1) harmful; (2) relevant to application domains of interest. I understand the narrative of the study, but I fail to depict scenarios where the identification of such biases is a mandatory requirement. For instance, should we consider some de-biasing process?
Could the authors clarify the motivations and impact of proposed study?
Overall, I don't see any strong inconsistency in the work. For this reasons, I'm more inclined to recommend acceptance. Further discussion on my concern would be highly appreciated to increase my score.

## Comments

[Table 2, Table 3] Could the authors report the standard deviation values as well?

## Typos and Writing

[Section 2.2, Line 117] 'the first:w PCA dimension'

## References

[1] Amandalynne Paullada, Inioluwa Deborah Raji, Emily M. Bender, Emily Denton, Alex Hanna:
Data and its (dis)contents: A survey of dataset development and use in machine learning research. Patterns, 2021

**Reproducibility:**

4: Could mostly reproduce the results, but there may be some variation because of sample variance or minor variations in their interpretation of the protocol or method.

**Reviewer Confidence:**

3: Pretty sure, but there's a chance I missed something. Although I have a good feel for this area in general, I did not carefully check the paper's details, e.g., the math, experimental design, or novelty.

---

> ### Author Rebuttal · Authors · 2023-08-24
>
> Our initial motivation was problematic NLP applications that evaluate individuals based on their CVs or records found online. We viewed our study as exploratory; we did not actively search for use cases where the discovered discovered could potentially be harmful. However, one example we found was that using the Distilled mUSE in such a sensitive (and inherently problematic) application, being from a poorer country, might have a similar effect as working in low-profile jobs.
>
> Thank you for mentioning the work of Paullada et al., which is very interesting, and we were unaware of it. There are indeed implicit normative assumptions when preparing the data we work with. We consider countries' geographical and political features present in the Western mainstream discourse, which are not necessarily the most appropriate in all contexts. Paullada et al. talk about benchmarking datasets, which are costly to produce, which motivates the community to reuse the datasets often outside of their original context. Designing templates is cheap compared to collecting a dataset from annotators. Also, given the exploratory nature of our study, we do not provide a benchmark others can compete on. Hence, the risk of problematic use of our data outside of the original context is relatively small.

---

### Meta-Review · Area_Chair_UJPG · 2023-09-23

**Recommendation:** 3

**Metareview:**

Summary (adapted from reviewer os6q): This is an exploratory paper that studies and exposes how pre-trained multilingual language models inherently contain bias or make distinctions among European countries. The authors show a reflection of geopolitics (Eastern vs Western European countries) and economy based (in terms of GDP) distinctions. Prestige related to jobs is also explored and is found to generally not correlate between country-related prestige.

The reviewers found this study to be generally sound (3/3/3) and acknowledged that it covers an interesting and novel topic.

Two reviewers raised concerns that the stereotypes explored in this work may prove harmful. This was well-addressed in the limitations and ethical considerations section of the paper.

There was some disagreement about the ease with which the paper could be understood, with one reviewer citing difficulty, one reviewer praising how well-written it is, and the other not mentioning it at all. The reviews suggest some areas that would benefit from expansion of the paper into 5 pages if accepted.

Overall, this paper provides an interesting and unique contribution, even if it addresses just two small questions about the biases of these models.

---

### Decision · Program_Chairs · 2023-10-07

**Decision:**

Accept-Findings

**Comment:**

Summary (adapted from reviewer os6q): This is an exploratory paper that studies and exposes how pre-trained multilingual language models inherently contain bias or make distinctions among European countries. The authors show a reflection of geopolitics (Eastern vs Western European countries) and economy based (in terms of GDP) distinctions. Prestige related to jobs is also explored and is found to generally not correlate between country-related prestige.

The reviewers found this study to be generally sound (3/3/3) and acknowledged that it covers an interesting and novel topic.

Two reviewers raised concerns that the stereotypes explored in this work may prove harmful. This was well-addressed in the limitations and ethical considerations section of the paper.

There was some disagreement about the ease with which the paper could be understood, with one reviewer citing difficulty, one reviewer praising how well-written it is, and the other not mentioning it at all. The reviews suggest some areas that would benefit from expansion of the paper into 5 pages if accepted.

Overall, this paper provides an interesting and unique contribution, even if it addresses just two small questions about the biases of these models.